# Field Chemical Immobilization of Free-Ranging Crested Porcupines with Zoletil^®^: A Reviewed Dosage

**DOI:** 10.3390/vetsci7040194

**Published:** 2020-12-01

**Authors:** Francesca Coppola, Enrico D’Addio, Lucia Casini, Simona Sagona, Antonio Felicioli

**Affiliations:** 1Departement of Veterinary Sciences, University of Pisa, Viale delle Piagge 2, 56124 Pisa, Italy; francesca.coppola@vet.unipi.it (F.C.); lucia.casini@unipi.it (L.C.); simonasagona@unipi.it (S.S.); 2Freelance Practising Veterinary Surgeon, Serravezza, 55045 Lucca, Italy; dadvet74@gmail.com; 3Departement of Pharmacy, University of Pisa, Via Bonanno 6, 56126 Pisa, Italy

**Keywords:** immobilization, anesthetic, tiletamine-zolazepam mixture, *Hystrix cristata*, rodent

## Abstract

The tiletamine-zolazepam mixture is a widely used anesthetic for chemical immobilization of wild mammals due to its short induction time, good muscle relaxation, smooth recovery with low convulsions occurrence, and minimal effect on respiration. An injection dose of 7–8 mg/kg of tiletamine-zolazepam has been proven to be an effective and safe immobilizing mixture for crested porcupines under field conditions. However, the occurrence of long immobilization and recovery times, with high excitement and convulsion during awakening, were recorded. In order to reduce such side effects after recovery, the effectiveness of a lower dosage (4–6 mg/kg) of tiletamine-zolazepam (Zoletil^®^) was tested. The results obtained confirm that the use of tiletamine-zolazepam in crested porcupine immobilization provides a quick induction, wide safety margin, and predictable awakening under field conditions. A smaller injection dosage of 5 mg/kg has been proven to be sufficient to ensure a short induction time (average: 7.1 min), with good muscle relaxation and little excitement of the animals during awakening. The lower dosage of tiletamine-zolazepam, while providing a shorter recovery time (average: 53.6 min), proves to be adequate for standard handling procedures. Furthermore, the smaller amount of tiletamine-zolazepam also ensures safe immobilization for pregnant individuals and porcupettes.

## 1. Introduction

Zoletilt^®^ 100 (Virbac, Carros, France) is an injectable anesthetic consisting of a mixture of tiletamine and zolazepam in an equal ratio (250 mg each). Tiletamine is a dissociative agent that produces analgesia and anesthesia [1]. Zolazepam is a muscle-relaxing benzodiazepine that contrasts the convulsive seizures associated with tiletamine [1]. Chemical immobilization is a common practice for the management and handling of wildlife [1,2]. The tiletamine-zolazepam mixture has been widely used for chemical immobilization of wild mammals, including rodents [3,4,5,6,7,8], due to its short induction time, good muscle relaxation, smooth recovery with few convulsions, and minimal effect on respiration [1].

The degree and duration of immobilization by tiletamine-zolazepam is dose-related, with wide safety margins [9]; in some species, ambient temperature can influence the dosage necessary for immobilization [10]. Exceeding doses of tiletamine-zolazepam are well tolerated by most species, which is useful when animal body weight can only be roughly estimated [11,12]. Moreover, the tiletamine-zolazepam mixture provides gradual and predictable animal recovery, which is helpful when handling potentially dangerous species [11].

The crested porcupine (*Hystrix cristata*) is the biggest Italian rodent [13,14]. It is monogamous and lives in burrows in family groups [15,16]. It is mainly herbivorous and mainly nocturnal [17,18]. The back, rump, and hips of the porcupine body are covered in long erectile quills functioning as defence–offense weapons [19,20]. Crested porcupine was recently identified as a new potential host for some zoonotic diseases such as giardiasis and leptospirosis [21,22,23]. This evidence necessitates further investigation to assess the general health status of this rodent in Italy. Massolo et al. [24] have proven that tiletamine-zolazepam mixture is an effective and safe immobilizing drug for crested porcupine under field conditions. Massolo et al. [24] recommended a Zoletil^®^ 100 injection dose for porcupine immobilization of 7–8 mg/kg. However, for this dosage, Dari [25] and Coppola [26] reported prolonged recovery times, exceeding 2 h, and convulsions during awakening in two distinct porcupines, indicating that the dosage used by Massolo et al. [24] may have to be revised. Chemical immobilization of captured individuals is an essential procedure in the monitoring of free-ranging crested porcupines’ health status and management. The aim of this investigation is to optimize the administration of tiletamine-zolazepam in order to reduce the recovery time and excitement during awakening. We predicted that a lower dosage of tiletamine-zolazepam is sufficient to ensure rapid induction time, lower but adequate recovery time, and lower excitement during awakening.

## 2. Materials and Methods

The capture-marking activity of resident porcupines was approved by the Italian Institute for Environmental Protection and Research (ISPRA) with protocol number 22,584 of 8 May 2017 and protocol number 150,071 of 16 March 2018 and by the Tuscany Region with Decrees n. 14,235 of 3 October 2017 and n. 4842 of 6 April 2018. 

Between 2018 and 2020, porcupine-capture campaigns were performed within a wider investigation on crested porcupine ecology and health status. The investigation was carried out in two experimental sites in the Province of Pisa: a hilly area in Crespina-Lorenzana (43.57181 latitude–10.55348 longitude) and the wildlife hunting reserve Camugliano (43.60210 latitude–10.64742 longitude). Porcupines were captured in twelve double-entrance box traps (110 × 42 × 42 cm) baited with corn and monitored by camera-traps. Capture traps were placed along pathways where signs of porcupine presence (i.e., quills, feces, and footprints) were regularly detected, and each trap was checked two times/day. The weight of each capture trap was recorded before the beginning of the capture procedures. Each captured porcupine was weighed, anesthetized, and sexed, and the age class was estimated on the basis of animal weight [17]. The total weight of the trap and porcupine was recorded by an electronic dynamometer, and the animal weight was obtained by difference. Porcupines were immobilized in the capture cage by intramuscular injections in the lumbar region with Zoletil 100^®^ (Virbac, Carros, France; 250 mg tiletamine as hydrochloride and 250 mg zolazepam as hydrochloride) using an air-compressed syringe (Mini-ject 2000, 2 mL, Alpha, San Miniato, Italy) administered by a blow-pipe of 1 m long. Syringes were checked after removing it from the animal body to assess full discharge. Porcupines captured in 2018 were immobilized using the injection dose of 7–8 mg/kg, as suggested by Massolo et al. [24]. In 2019 and 2020, the injection dose was 4–6 mg/kg. For each captured porcupine, the induction time and recovery time were recorded. Induction time is defined as the time from the injection until the animal’s lack of reaction to manhandling. The resting of the head on the ground was considered the beginning of the state of immobilization. Recovery time is defined as the time from induction to the time in which the animal shows a reaction to external stimuli and an upright posture. All individuals were released once they had recovered. The handling procedures included clinical examination (i.e., visual inspection of teeth, skin, and all mucous membranes and palpation of the abdomen), recording of initial body temperature and heart rate, collection of blood and saliva samples, and application of an individually marking with colored adhesive tapes on the quills or white or black paint sprayed on the body.

## 3. Results

Overall, 18 porcupines were captured and immobilized, of which there were 6 adult females, 5 adult males, 4 subadult females, 1 subadult male, and 2 female porcupettes (Table 1); 27.8% (*n* = 5) of porcupines were captured in 2018, 44.4% (*n* = 8) in 2019, and 27.8% (*n* = 5) in 2020. No porcupines died after chemical immobilization, and a captured pregnant female successfully gave birth to two porcupettes 23 days after immobilization. The average initial body temperature was 38.4 °C in porcupines captured in 2018, and 38.1 °C in 2019–2020; no cases of hyper- or hypothermia were recorded. Initial heart rate after immobilization was similar in porcupines captured in 2018 (average: 117.8 bpm) and those captured in 2019–2020 (average: 117.4 bpm). In 2018, the average tiletamine-zolazepam injection dose was 7.6 mg/kg, while in 2019–2020, was 5.0 mg/kg. All darts were completely discharged after removal from the animal’s body. Multiple injections were necessary in 2/18 (11.1%) cases to obtain a suitable immobilization time that was necessary to treat pre-existing injuries. In both cases, a supplementary dose of 2 mg/kg was administered. The average induction time was 13.0 min in 2018 and 7.1 min in 2019–2020, while the average recovery time in 2018 was 103.4 min, and in 2019–2020, 53.6 min (Table 1). All immobilized animals showed good muscle relaxation and no excess of salivation during immobilization. Excitement and increase of motor activity after recovery were recorded mainly in porcupines immobilized in 2018.

## 4. Discussion

The results obtained in this investigation confirm that tiletamine-zolazepam in crested porcupine immobilization provides a quick induction, wide safety margin, and manageable recovery process under field conditions, as proven by Massolo et al. [24].

In this investigation, the injection dosage used for crested porcupine immobilization in 2018 was 7–8 mg/kg, as suggested by Massolo et al. [24], while the dose ranged from 4 to 6 mg/kg in captured porcupines in 2019–2020. The average induction time in porcupines immobilized in 2018 (average: 13.0 min) was greater than those obtained in 2019–2020 (average: 7.1 min) and those reported by Massolo et al. [24] (average: 5.3 min). It is noticeable that in 2018, in three out of five captured porcupines, the induction time ranged between 15 to 20 min, while for the other two individuals, the induction time was similar to those obtained in 2019–2020 and by Massolo et al. [24]. Therefore, the discrepancies in the results obtained both in this study and between this study and Massolo et al. [24] could be due to incomplete intramuscular discharge of dart due to the lateral position of the syringe injection hole. Furthermore, this result could also be due to different body conditions among individuals as well as health status or stress levels following the capture event.

The average recovery time of porcupines immobilized in 2018 (average: 103.4 min) were higher than those obtained in 2019–2020 (average: 53.6 min) when using a lower tiletamine-zolazepam dosage. Moreover, the porcupines’ average recovery time in 2018 was higher than those reported by Massolo et al. [24] (average: 28.7 min) using the same tiletamine-zolazepam dosage. Additionally, in this case, variation in the recovery time could be due to different body conditions and health status of the captured individuals as well as stress levels following the capture event. In contrast to Massolo et al. [25], in this investigation, no miscarriage was recorded in the porcupine adult pregnant female immobilized in 2019–2020. This result suggests that the use of a lower tiletamine-zolazepam dose did not have an impact on pregnant individuals. The absence of tiletamine-zolazepam impact on pregnant individuals was also recorded by Lariviere and Messier [3] in striped skunks and Ballard et al. [27] in grey wolf. Results obtained in this investigation also clearly show that the use of a lower injection dosage of tiletamine-zolazepam determines a reduction of recovery time as well as excitation state after awakening in porcupines. Moreover, injection of a smaller dose can reduce the risk of tissue damage due to injection under pressure of the solution [28].

## 5. Conclusions

In conclusion, a single intramuscular injection dose of 4–6 mg/kg of tiletamine-zolazepam mixture is enough to ensure a short but adequate porcupine immobilization time for most handling procedures, with good muscle relaxation, few convulsions after awakening, and a safe immobilization of pregnant female and porcupettes. The authors are aware of the small sample size, and further investigation is needed in order to enlarge the sample size, ensure complete intramuscular discharge of the dart, and assess the relationship between the effects of the anesthesia on animals and their health status.

## Figures and Tables

**Table 1 vetsci-07-00194-t001:** Mean value and standard deviation of BW, BT, HR, IjD, InT, and RT of porcupines captured in 2018 and 2019–2020, respectively.

Sample	Age Class	Sex	BW (kg)	BT (°C)	HR (bpm)	IjD (mg/kg)	InT (min)	RT (min)	Clinical Status
**2018**									
**1**	A	F	13.5	37.8	115	7	20	100	Good
**2**	A	F	11.0	38.1	119	7	15	105	Good
**3**	SA	F	5.0	39.0	121	8	6	101	Good
**4**	A	M	11.0	38.6	118	8	17	118	Good
**5**	A	M	14.5	38.4	116	8	7	93	Good
Average ± SD			11.0 ± 3.7	38.4 ± 0.5	117.8 ± 2.4	7.6 ± 0.5	13.0 ± 5.6	103.4 ± 9.2	
**2019–2020**									
**6**	P	F	3.5	38.6	120	4	4	34	Good
**7**	SA	F	7.2	37.9	121	5	6	34	Good
**8**	A	M	11.0	38.3	116	6	10	43	Severe injury on the nose
**9**	A	F	12.0	38.4	117	5	4	56	Good
**10**	A	M	14.5	38.7	116	5	10	48	Good
**11**	A	F	11.0	38.5	119	5	7	53	Good
**12**	P	F	1.9	38.7	123	4	2	43	Good
**13**	SA	F	9.5	37.9	114	5	4	86	Good
**14**	A	M	11.9	36.0	110	6	8	67	Severe injury in the rump
**15**	SA	F	9.4	37.2	112	5	10	60	Good
**16**	SA	M	10.6	38.2	113	5	8	57	Good
**17**	A	F	14.7	38.2	129	5	8	62	Pregnant
**18**	A	F	15.2	38.5	116	5	11	54	Lactation
Average ± SD			10.2 ± 4.0	38.1 ± 0.8	117.4 ± 5.1	5.0 ± 0.6	7.1 ± 2.9	53.6 ± 14.1	

Age class (A: adult, SA: subadult, P: porcupette), sex (M: male, F: female), body weight (BW), body temperature (BT), and heart rate (HR) of each immobilized porcupine in 2018 and 2019–2020. For each individual, tiletamine-zolazepam (Zoletil^®^ 100) injection dose (IjD), induction time (InT), recovery time (RT), and clinical status are reported.

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
