# Peer review of "Field Chemical Immobilization of Free-Ranging Crested Porcupines with Zoletil®: A Reviewed Dosage"

_vetsci, 2020, doi:10.3390/vetsci7040194_

Round 1
Reviewer 1 Report
N/A
Author Response
Authors wish to thank the Reviewer 1 for its positive judgements about the manuscript
Reviewer 2 Report
English needs correcting/improving throughout
Introduction: include references where Zoletil has been used in rodents eg King 2010 (Zoo Biology)
line 31: zolazepam is a benzodiazepine not a barbiturate
lines 41-43: superfluous references here - not relevant and should be reduced or removed
lines 48-50: this statement is confusing since Massolo et al do not mention convulsions on recovery and mean recovery time was reported as 28.7 minutes (although data from only 6 animals is reported), compared with 53.6 minutes in the current study. More explanation is therefore needed if the reports of long recovery times and convulsions are from unpublished theses (ref 26 and 27) - for example state that 'Massolo et al recommend a dose of 7-8mg/kg, however Dari and Coppola found that this dose range resulted in prolonged recoveries exceeding 2 hours and convulsions on recovery' Explain how many animals were anaesthetised in these studies compared to the study by Massolo et al.
line 66: weighed not weighted
line 71: did all darts discharge 100% ?
line 76-77: this seems a strange definition of recovery. would make more sense to define it as reaction to external stimuli and an upright posture - and then to state that animals were released once they had recovered. This would also be consistent with Massolo et al.
line 98: reduction of salivation compared with what? Do you mean there was no excess salivation?
line 107: I'm not sure than recovery can be described as predictable when it ranges from 118 minutes in this study to 18 minutes in Massolo et al.
line 109: similarly the absence of miscarriage in one pregnant female does not confirm that there is no impact on pregnancy, especially when one female aborted in Massolo et al.
line 114: instead of 'resulted higher' say 'was greater than'
line 115: further discussion is needed on this inconsistency. What body condition were the 2 groups of animals in? Based on your examination and any blood tests, were the animals healthy or not? If no evidence that there was a difference in health status what else could explain the discrepancy in results, both in this study and between this study and Massolo et al? Was there are difference in darting technique? Can you be sure that all darts discharged 100%?
Reviewer 3 Report
I think it's a well-articulated job. I would recommend adding references on the influence of temperature in the final discussions or introduction as reported in this recent article:
-Influence of Ambient Temperature and Confinement on the Chemical Immobilization of Fallow Deer ( Dama dama ) J Wildl Dis . 2017 Apr;53(2):364-367. doi: 10.7589/2016-06-131. Epub 2017 Jan 25.
Reviewer 4 Report
This research shows compares the efficiency of different dosages of Zoletil to immobilize crested porcupines as well as their side-effects on the animals.
- This reviewer have doubts about the novelty of this research. For example, is the use of zoletil authorized in wild animals?
- SD values should be included in table 1 instead in the manuscript body with the following format 39.4±0.46,….
Author Response
Response to Reviewer 4 Comments
Ms. Title “Field chemical immobilization of free-ranging crested porcupines with Zoletil®: a reviewed dosage”.
Submission ID: Vetsci-985404
Authors wish to thank the Reviewer 4 for its suggestions. All changes have been incorporate in the manuscript and detailed point-by-point response to each specific comment has been given below. All changes are tracked in the manuscript.
Reviewer comments:
This research shows compares the efficiency of different dosages of Zoletil to immobilize crested porcupines as well as their side-effects on the animals.
Point 1: This reviewer have doubts about the novelty of this research. For example, is the use of zoletil authorized in wild animals?
Response 1: According to the directive 2004/28 /CE and Legislative Decree n. 193/2006, since no drug anaesthetic registered for wildlife species is available we can use any other drug authorized for animal use even if for other animals species. Zoletil, is a veterinary anaesthetic drug authorized for cats and dogs and then, according to the law reported above, can be use also for wildlife. This fact cannot be considered a novelty of this manuscript while a little contribution to novelty could be considered the reduction of the dosage used for porcupine at the light of the single published paper concerning anaesthesia of porcupine with Zoletil. This topic has now been better outlined in the manuscript from line 58 to 67.
Point 2: SD values should be included in table 1 instead in the manuscript body with the following format 39.4±0.46,….
Response 2: The SD values has been removed from the text and has been included in table 1 using the format suggested by the reviewer

Round 2
Reviewer 4 Report
After reviewing the revised version of the manuscript, the doubts were solved and the manuscript was properly improved. Thus, in my judgment, the manuscript is suitable for its publication.